# The PRISMA 2020 Statement: A System Review of Hospital Preparedness for Bioterrorism Events

**DOI:** 10.3390/ijerph192316257

**Published:** 2022-12-05

**Authors:** Lulu Yao, Yongzhong Zhang, Chao Zhao, Feida Zhao, Song Bai

**Affiliations:** 1Emergency Medicine, Institute of Disaster and Emergency Medicine, Tianjin University, Tianjin 300072, China; 2Epidemiology and Health Statistics, Institute of Disaster and Emergency Medicine, Tianjin University, Tianjin 300072, China; 3Center for Biosafety Research and Strategy, Tianjin University, Tianjin 300072, China; 4Institute of Disaster and Emergency Medicine, Tianjin University, Tianjin 300072, China; 5Evaluation and Optimization of Health Emergency Response Capacity, SD, Institute of Disaster and Emergency Medicine, Tianjin University, Tianjin 300072, China

**Keywords:** bioterrorism, hospital preparedness, disaster, public health

## Abstract

Hospitals are an important part of a nation’s response to bioterrorism events. At present, research in this field is still in the initial stage. The number of related studies is small, the research direction is relatively concentrated, and a comprehensive analysis and standard evaluation system are lacking. This literature survey was conducted using PRISMA methodology. Collective information was gathered from PubMed, Web of Science, Scopus, and available grey literature sourced through Google and relevant websites. The studies were screened according to the Preferred Reporting Items for Systematic Reviews and Meta-Analysis (PRISMA) flowchart. Analysis and summary of the extracted data was performed according to the World Health Organization (WHO) Rapid Hospital Readiness Checklist (2020). Twenty-three articles were selected for review, data extraction, and data analysis. Referring to the WHO rapid hospital readiness checklist, six main indicator categories were determined, including emergency management, medical service capacity, surge capacity, laboratories, regional coordination, and logistical support, and fifty-two subcategories were finally identified. The study summarizes and analyzes the relevant literature on hospital disaster preparedness and extracts relevant capability elements, providing a reference for the preparation of hospitals against bioterrorism events and a basis for the design and development of hospital preparedness assessment indicators.

## 1. Introduction

Bioterrorism refers to the intentional release of viruses, bacteria, toxins, or fungi to cause panic, mass casualties, or serious economic damage [1], with latent, sporadic, contagious, panic-related, and catastrophic characteristics. Bioterrorism is an important manifestation of terrorism, and it may cause physical and psychological harm, generate socially negative public opinion, serious social panic, and other derivative harm [2]. From 1979 to 2021, more than 40 bioterrorism cases were reported worldwide [1], targeting governments and citizens and attracting widespread attention [3].

Countries around the world have successively issued policies and laws to deal with bioterrorism [4]. The World Health Organization (WHO) has also issued a hospital readiness checklist to promote disaster preparedness. The International Criminal Police Organization (Interpol) made it clear at the International Conference Against Bioterrorism in Lyon, France in 2005 that bioterrorism has become one of the biggest security threats in the world. In 2021, China promulgated the “Biosecurity Law”, which emphasized that traditional biosecurity issues and new types of biosecurity risks are superimposed and that biosecurity criminal activities have seriously threatened the safety of human life. Therefore, the ability to quickly issue and deploy a response in reaction to sudden bioterrorism events is an important issue that urgently needs to be solved.

Hospitals are an integral part of the entire bioterrorism emergency management system [5], as they are the first places to receive patients who have been affected by such attacks. The Sendai Framework for Disaster Risk Reduction 2015–2030, adopted by 187 countries, explicitly emphasizes the need to strengthen the resilience of hospitals to disaster risks [6]. Hospital bioterrorism preparedness refers to the ability to rapidly respond and deploy resources in reaction to emergencies, including real-time epidemiological monitoring of bioterrorism factors, organizational management of hospitals, diagnosis and treatment of patients, and surge capacity, as well as preparedness, prehospital first aid, treatment, and recovery [7]. When a bioterrorism event occurs, a hospital’s emergency management response, communication between superiors and subordinates, coordination of various departments, medical staff capabilities, material reserves, and infection control all face challenges.

Since the outbreak of COVID-19, the number of research articles about bioterrorism has gradually increased and the preparedness of hospitals to bioterrorism events has become a focus of attention. Nevertheless, the field of study is in its infancy and the research orientation is relatively narrow, with most studies focusing on biological warfare agents, pathogens, and other clinical aspects. A comprehensive analysis and standard evaluation system to test the ability of hospitals to respond to bioterrorism are also rare. The purpose of this study is to summarize and analyze the relevant literature on hospital preparedness for bioterrorism events and extract the relevant capability elements, thus providing a reference for the preparation of hospitals against bioterrorism and a basis for the design and development of hospital preparedness assessment indicators.

## 2. Methods

This review was performed in accordance with PRISMA guidelines.

### 2.1. Search Strategies and Databases

The study searched PubMed, Web of Science, and Scopus, using a combination of the following search terms (in Title/Abstract): “hospital”, “emergency trauma”, “bioterrorism”, “emergency”, “emergency preparedness,” and “disaster planning”. The systematic search strategy is shown in Table 1. All relevant studies published from 1 January 2012 to 1 September 2022 were selected, including white papers and conference documents. This review did not limit the included research design methods and used an advanced keyword combination strategy to search for articles.

### 2.2. Selection Process and Eligibility Criteria

The selected articles were imported into EndNote, which intelligently removed any duplicate articles. The authors sorted the titles alphabetically and manually deleted duplicates to prevent the omission of deletions due to missing titles and different default articles in EndNote.

This study selected original studies using the eligibility criteria given in Table 2. The authors recorded high-frequency words that appeared in the literature abstracts for research reference. All articles deemed eligible were successfully downloaded. 

### 2.3. Quality Assessment

The study used the PRISMA 2020 checklist to screen articles, extract information, and summarize relevant indicators based on the WHO Rapid Hospital Readiness Checklist (2020). The study performed a descriptive synthesis and analysis of the data, but due to the limited number of studies and heterogeneity of the data, the study did not perform a meta-analysis. The reviewers comprised 4 people, 2 of whom have doctor’s degrees and 2 of whom have master’s degrees. All reviewers have been engaged in research investigating social medicine and health service management for a long time, especially research exploring large-scale population aggregation.

## 3. Results

A total of 2139 records were initially scoped in the literature search, 955 of which were duplicates. After screening for eligibility and inclusion criteria, 23 articles were ultimately included in the study (Figure 1).

### 3.1. Quantitative Analysis

According to the PRISMA descriptions (Table 3), most of the 23 studies (*n* = 16) were cross-sectional surveys, while others were Delphi studies (*n* = 4) and systematic reviews (*n* = 3). Most of the articles were surveys of mass casualty/bioterrorism response capabilities or retrospective analyses of past events. For the study region, Iran published the most articles (*n* = 10), followed by China (*n* = 4), Saudi Arabia (*n* = 2). This suggests that research on bioterrorism is closely related to the local political environment. The majority of the literature in this review was published between 2020 and 2021 (*n* = 7), possibly due to the COVID-19 pandemic, which was concentrated during this period. Overall, the indications emphasized that bioterrorism has increased in recent years and that the preparedness of hospitals to deal with bioterrorism events should be improved.

### 3.2. Qualitative Analysis

Using the PRISMA 2020 checklist, the authors combed out the objectives, research methods, survey participants, response rate, index of composition, discussion, and limitations of the 23 selected articles, and extracted 52 factors, but found that they may overlap in one or more domains. In 2020, the WHO released the Rapid Hospital Readiness Checklist, arguing that hospital emergency preparedness includes 12 critical components, including leadership and event management systems, coordination and communication, monitoring and information management, risk communication and community engagement, management, financial and business continuity, human resources, surge capacity, continuity of basic support services, patient management, occupational health, mental health and psychosocial support, rapid identification and diagnosis, and infection prevention and control [8]. Using these components as a reference, we organized and summarized the extracted factors after discussion, finally dividing the 52 factors into 6 main categories, summarized as 15 indicators.

The major categories included emergency management (bioterrorism response, risk communication, training and drills, leadership), medical service capabilities (monitoring and warning, emergency medical services, patient management, infection control, cadaveric disposal), surge capacity (staff, equipment, pharmaceuticals), laboratories, regional coordination, and logistical support. Extracted categories such as transport, agent identification, and in-hospital standard operating procedures were included in these categories. Table 4 shows the separate factors that were extracted from the articles on hospital preparation for biological events.

#### 3.2.1. Emergency Management

Emergency management is one of the main criteria for hospital preparedness. For emergency management, bioterrorism response, risk communication, training and drills, and organization and coordination were more frequently considered, mentioned in 9, 10, 13, and 6 articles, respectively, whereas transport and information management were considered only in 1 and 2 studies, respectively.

##### Bioterrorism Response

Bioterrorism response refers to the urgent action taken before, during, and after a bioterrorism event with a view to reducing casualties, damage, and disruptions [9]. According to the findings, it was generally accepted that standard operating procedures (SOPs) are essential requirements for an emergency response [10]. Therefore, hospitals can develop a detailed plan to help medical personnel reach a state of emergency faster.

##### Risk Communication

Risk communication refers to the process of information exchange and meaning sharing in health emergency or emergency situations, including the “real-time exchange of information, advice and opinions” between leaders, experts, and the public [11]. Communication is considered to be one of the main challenges in regard to emergency management during biological events [12]. 

##### Training and Drills

Training and drills are important ways to validate hospital preparedness plans and identify potential problems. They are mainly simultaneous and comprehensive tests of emergency planning, staffing levels, personnel training, procedures, facilities, equipment and materials, and risk assessment [13].

##### Leadership

Leadership in crisis is one of the important administrative behaviors and key to emergency management. This factor was also referred to in some studies as command and coordination. Leadership functions cover the assessment of transmission patterns, patient conditions, appropriate staffing levels, the supply of personal protective equipment, and the use of medical resources.

#### 3.2.2. Medical Service Capacity

A hospital’s medical service capacity and medical level are important factors in their response against bioterrorism. For medical service capacity, triage, personal protective equipment (PPE), monitoring and warning, and decontamination were identified as essential in 5, 6, 6, and 8 studies, respectively. Electronic medical records, treatment and infection control, and body storage/disposal were the subjects of few studies, only mentioned in one or two studies.

##### Monitoring and Warning

Monitoring and warning is an important link in discovering bioterrorism [14]. The early monitoring of biological diseases can effectively alert officials to the presence of potential bioterrorism events. 

##### Emergency Medical Services

The emergency medical services provided by hospitals are important for all disasters, and may be divided into two parts: prehospital and in-hospital. Prehospital services are primarily based on triage and performed by medical staff with a background in anaesthesia, critical care, and emergency or recovery medicine. These services require quick judgment and first aid for injuries, and predetermine patient destinations according to the characteristics of their injuries and hospital treatment characteristics. 

##### Patient Management

Patient management includes admission or referral, triage, diagnosis, treatment, patient flow and tracking, discharge, and follow up, as well as the management of support services, pharmacy services, and logistics and supply functions [8]. It is an important factor affecting survival and mortality rates, and also the basis for judging the level of service provided by a hospital.

##### Prevention and Control of Infections

Biologically infectious diseases are highly contagious and all patients admitted due to a biologically infectious disease may carry the source of infection. Therefore, prevention and control of infections are priorities of hospitals. Generally, hospitals can effectively prevent medical staff or infrastructure from exposure to toxic substances and secondary infections by being equipped with decontamination and personal protective equipment and by setting up quarantine areas.

##### Corpse Disposal

Death management is the embodiment of a hospital’s service capacity. Bioterrorism is mostly associated with infectious or radiation-borne diseases, and the death toll related to these events can be quite high, leading to the production of large numbers of bodies that may be infectious [15]. This may pose a threat to all hospital staff; therefore, proper disposal of bodies is very important. 

#### 3.2.3. Surge Capacity

Surge capacity is the ability of a hospital to expand its resources to deal with a large number of casualties without external support [16]. For surge capacity, infrastructure and equipment, pharmaceuticals, human resources, and surge bed capacity were the most commonly discussed factors in 10, 6, 8, and 8 studies, respectively. Psychological counselling was only considered in one study.

##### Human Resources

One of the most important parts of a surge plan is to increase the number of employees or human resources [17]. The influx of large numbers of patients can present significant challenges for health care workers. In this case, it is often necessary to immediately activate the personnel recall mechanism, call in volunteers, and replace personnel roles and responsibilities in emergency situations. 

##### Medical Equipment

During bioterrorism events, the use of biological warfare agents can lead to an explosive increase in the number of patients, which can result in the resources required by hospitals and emergency departments to far exceed those required for daily operations [18]. Medical equipment is an important manifestation of the surge capacity of hospitals. It is necessary to reserve a certain number of hospital beds, oxygen cylinders, ECG monitors, ventilators, disinfection machines, and other basic medical equipment related to biological diseases [19]. 

##### Pharmaceutical Reserve

Stockpiles of drug supplies needed to respond to bioterrorism attacks include appropriate antibiotics, vaccines, atropine, and antidotes for common nerve agents. Regardless of the quality of the response design, hospital systems come to a standstill when supplies of medicines and other materials are exhausted [5]. 

#### 3.2.4. Laboratories

For laboratories, scientific research and surveillance systems were the most concerning factors, which were mentioned in three publications. Identifying the type of bioterrorism agent plays an important role in improving the response of hospitals to bioterrorism events [20]. 

#### 3.2.5. Regional Coordination

For regional coordination, integration with local or regional systems was obviously an important aspect, which was considered in three reports. Cooperative planning and health facility networking were only mentioned in one literature study.

#### 3.2.6. Logistical Support

For logistical support, funding was considered the most important factor, identified in 7 studies. Architectural and lifeline facilities were also proposed by 4 and 6 studies, respectively. Legal and waste management were not common factors, only considered in one article. Logistical support is an important type of support needed to help hospitals deal with a bioterrorism event, mainly involving functional financial and security support. Funding plays a decisive role and directly affects the capacity of medical services. Scholars have argued that it is necessary to evaluate the logistics of hospitals. During an emergency or disaster, the normal operation of hospital functional facilities will ensure that it can continue to provide health services when such services are most needed [21].

## 4. Discussion

In recent years, hospital preparedness for bioterrorism events has attracted more attention. This PRISMA review included 23 peer-reviewed articles published from 2012 to 2022, and the main conclusions of each study were extracted and summarized. Overall, bioterrorism response, risk communication, training and drills, and organization and coordination were often considered to be the main components of emergency management, while out-of-hospital aspects, such as transport, were less valued. For medical service capacity, triage, personal protection equipment, monitoring and warning, and decontamination were often regarded the main criteria, while electronic medical records and corpse disposal were less commonly considered. In terms of surge capacity, medical equipment, pharmaceuticals, human resources, and the number of hospital beds were noted as major categories, extensions of which became other factors. Laboratories, regional coordination, and logistical support were also frequently regarded, while waste management and body disposal were given a lower priority in most articles.

### 4.1. Emergency Management

This study found that bioterrorism response preparedness can help hospitals to maintain essential services and normal operation, reduce the damage caused by insufficient preparation or inexperience of hospital staff, and is an important means to achieve routine disaster preparedness. It was also proposed that hospitals should formulate plans to respond to bioterrorism based on specific criteria, which can reduce costs and effectively prevent and mitigate harm [22]. In addition, hospital response plans need to be updated regularly and shared with other organizations (EMS, emergency management agencies, local governments, police, and other hospitals) to clarify expectations for agency performance and identify the types of support shared between responding organizations [13]. 

Surprisingly, risk communication plays an important role in emergency management. Once a communication breakdown occurs, the network will be paralyzed and the internal and external information exchange will be terminated, thus affecting the operation of the whole hospital. At the same time, public communication is also necessary. Hospitals should release accurate and clear information in a timely manner to meet public information needs and guide public opinion, so as to reduce fear and anxiety in the community and prevent the spread of rumors [23]. This can also control the number of patients rushing to the hospital to a large extent, help the hospital respond quickly, improve the hospital’s strain capacity, and reduce the surge pressure. 

Training and drills can enable staff (doctors, nurses, support staff) to acquire the skills and knowledge required to deal with infections and biological events. Regular training with simulations and drills can avoid the false sense of security and the so called “paper plan syndrome”. It was argued that the training of health professionals in emergency departments can play a key role in managing victims of bioterrorism [24]. At present, the most commonly used method is desktop deduction, which can help reduce costs to a certain extent [25].

Research shows that leadership is valued in response to bioterrorism attacks. Risk prediction, judgment, and decision-making by leaders run through the entire reaction process and guide the function of command and coordination. Leaders should act as bridges to convey instructions to medical staff in a timely manner and simultaneously convey important information to the chain of command in order to ensure that the health system leadership understands the needs and perspectives of front-line workers [26].

### 4.2. Medical Service Capacity

Studies have shown that monitoring and warning systems employed in hospitals are unique. Unlike surveillance in other departments, hospitals can detect potential biological diseases earlier, establish a virus transmission model, and improve biosafety defense planning by analyzing the association between in-patient conditions over a period of time. In this way, precautions can be taken in advance to reduce the surge in hospitals.

Emergency medical services can help guide the allocation of limited medical resources [9]. Especially, prehospital triage can prevent the surge in patients. To minimize the negative impact on acutely ill and injured patients, increasingly sophisticated and evidence-based triage algorithms have been established as the standard of care [27]. A study also pointed out that a disaster medical assistance team (DMAT) has been established in the United States, providing a reference for the appropriate construction of a wholistic hospital departmental response [28].

Studies have found that the management and treatment of patients in hospitals are not standardized and sufficiently modernized. The first is due to the limited use of electronic medical records. The use of electronic records in hospitals could lead to greater monitoring capabilities, allowing hospitals to track emergency department admissions for influence-like illnesses and improve antibiotic prescription rates, providing better care for patients [29]. However, current electronic medical records do not link prehospital and in-hospital visitations, and there is no information sharing among hospitals. The second is due to the lack of classification schemes for biological diseases. Biological triage has a different mechanism from other triage methods, and an appropriate triage protocol is an important part of hospital emergency preparedness. However, the lack of applicable classification schemes will lead to the failure of high-priority treatment for infection transmission in triage populations; therefore, hospitals will not be able to provide services for most patients in a short time [23]. In addition, it is not only necessary to consider the injured individuals, but also to distinguish key populations who are vulnerable to biological infectious diseases, such as pregnant women, the elderly, and children, as well as the corresponding departments that should be prepared to deal with bioterrorism events.

Studies have shown that improving the vaccination rate of the whole population is an effective way to prevent biological infection. Surprisingly, this study showed a significant proportion of medical staff and the public were reluctant to participate in the current voluntary vaccination program. This may be because the event influenced healthcare workers’ perception of the vaccine [30]. Therefore, hospitals need to conduct vaccination work to ensure that all medical staff are vaccinated. 

Hospitals need to ensure that all patients are decontaminated before they enter the hospital. After biological events, many people arrive at the hospital in different ways, often bypassing on-site emergency medical treatment [31]. Thus, hospitals must provide PPE for health care workers. However, PPE is fairly simple and cheap for most medical personnel who treat victims of biological weapons. Reference should be made to standard precautions (Occupational Safety and Health Administration (OSHA) Class D), including gloves, gowns, hats, goggles, shoe covers, and N-100 or HEPA filtration masks [5]. In addition, hospitals should establish quarantine areas for infected patients to receive treatment services. The establishment of these isolation areas should refer to the idea of a square cabin when choosing a large space, where the entrances and exits should be equipped with sterilization devices, and internal and external patient transportation, placement, discharge, and autopsy care procedures should be carried out in strict accordance with the infection prevention policy. After the emergency phase is over, the entire hospital must be inspected to confirm that there are no lingering residual biological sources of infection and that contaminated pieces of equipment and materials have been disinfected or removed.

This study also found that morgue facilities and cadaver handling were neglected. The fatality rate of bioterrorist events is high, and during the peak period, bodies can pile up due to inadequate morgue facilities. This increases the risk of contagion. The COVID-19 outbreak has also highlighted the importance of ensuring hospital mortuary capacity. Therefore, adequate morgue capacity, temporary morgue space, refrigerated storage facilities, and trained staff are essential to ensure proper identification and disposal, and to reduce the risk of transmission.

### 4.3. Surge Capacity

This study found that human resources were the focus of hospitals, but vaccination, incentives/disincentives, and volunteer themes were ignored by most studies [32]. One survey showed increased willingness to participate when provided incentives, with 52% of respondents reporting a willingness to serve in the medical reserve corps in the event of a public health emergency and 72% reporting a willingness to care for patients with smallpox if they were guaranteed post-exposure vaccination (Mackie, 2022) [29]. Therefore, hospitals can improve employee motivation, morale, and satisfaction by offering incentives such as bonuses, child-care arrangements, volunteer networks, adequate training, and temporary housing, as well as subsistence support for health care workers and their families.

In addition, equipment and drug reserves also emerged as important factors in many studies. The COVID-19 pandemic has proven that stockpiling is the cornerstone of a holistic approach to disaster preparedness. When there is a surge in demand for additional patient populations, testing and treatment equipment is very important for staff and patients, affecting the level of treatment in hospitals [5]. Among these needs, bed capacity is the most relevant factor for emergency response planning and resource availability [33]; thus, hospitals need to plan emergency bed capacity in advance. The WHO also emphasizes that hospitals should have a stockpile of at least one week of adequate emergency medicines and supplies when preparing for disasters [34]. Hospitals also need to regularly review and dynamically use inventory to ensure that stored equipment and other items are effectively used before expiration.

### 4.4. Laboratories

It has been realized that hospital laboratories are an important part of tackling bioterrorism events. Hospitals must have complete laboratory identification and management procedures, with the ability to identify the type of bioterrorism agent, track and analyze all patient specimens and results, and collect and manage evidence. It is also necessary to prevent the occurrence of laboratory infections. Because of the infectious nature of biological terrorist agents, new infectious diseases may be caused during collection, labeling, packaging, handling, and internal and external transportation [35]. Therefore, laboratories should do a good job of protecting staff to avoid laboratory infections. 

### 4.5. Regional Coordination

Studies suggested that hospitals are single entities and their abilities are limited. Therefore, it is necessary to realize coordination of hospital work and regional mutual assistance to optimize the use of resources. It was proposed that regional grids should be implemented based on geographic locations, coordinated linkages, resource allocation, and mutual assistance, including patient overflow transport and community coordination between medical facilities and public safety [17]. In addition, attention should be paid to the coordination between local and central governments to ensure smooth communication and achieve information reciprocity.

### 4.6. Logistical Support

This study also found that logistical support was the basis for hospital responsiveness to bioterrorism. However, hospitals now face a “financial crisis” as they plan and prepare for bioterrorism events [36]. Whether in training or equipment, financial support is far from adequate. The financial support for hospitals by governments needs to be reasonably distributed according to the size of the hospital and actual needs. Private donations and fundraising by non-governmental organizations are also important sources of funding. 

The backup power supply was also neglected in most studies. Electricity is an important lifeline of a hospital. A survey conducted in Japan found that 65% of disaster relief hospitals believe power to be the most important lifeline for hospital operations. Hospitals must have a backup power source to ensure the continuance of intensive care units and operating rooms. Additionally, safety water tanks should be stocked to meet the hydration needs of staff and patients for at least three days [22], and firefighting facilities and safe passages should be regularly inspected and agreed upon with firefighting agencies. Fire alarm systems should be installed. The handling of bioterrorism viruses also requires that particular attention be paid to ventilation.

This research found that waste management was the least valued among logistical supports. However, waste management is an important part of hospital work, especially considering that medical waste contains hazardous substances such as infectious, toxic, and radioactive substances. Improper waste management can lead to secondary disasters that negatively affect humans and the environment. Therefore, there is a need to develop standardized procedures to manage hospital waste during bioterrorism events.

As highlighted in the above discussion, all of these less prioritized factors have a significant impact on bioterrorism preparedness. It is important to address these neglected aspects in the same way as the frequently prioritized areas.

## 5. Limitation

Due to the lack of attention to bioterrorism, some of the literature sourced was published earlier and the content was mixed. As a result, this study found deficiencies in the retrieval strategy when conducting the literature search. Further, the final results have not yet been validated in hospitals. 

## 6. Conclusions

In this literature review, the authors screened 23 studies published between 2012 and 2022 that were highly relevant to hospital disaster preparedness, including both quantitative and qualitative studies on hospital preparedness. Based on the WHO Rapid Hospital Readiness Checklist, this study analyzed and summarized the relevant capability elements of hospital bioterrorism preparedness, and discussed the role of key capabilities in hospital bioterrorism preparedness. The results show that a sound biological terrorism contingency plan should use existing resources, meet the demands of emergency coordination processes, play a key role in the effective response to the disaster, and assess hospital health facilities, as one of the most important factors of preparation. Another important consideration is the surge capacity of the hospital. Strengthening a hospital’s capacity in regard to patient management, human resources, material reserves, etc., has a positive effect on improving its coping capacity.

This study found that current studies in the literature in this area mostly used questionnaires and lacked the methods, tools, or comprehensive indicators for a systematic review. Each country and organization has expanded indicators and methods that fit their cultural backgrounds, but these tools have not been used on a large scale. These evaluation indicators and tools need to be further developed after undergoing a systematic evaluation and expert discussions based on WHO standards and national conditions.

The results of this study can be used as a reference standard for hospital preparation against bioterrorism and provide a basis for developing and designing hospital preparedness assessment indicators, thus helping hospitals respond to bioterrorism incidents more actively and effectively.

## Figures and Tables

**Figure 1 ijerph-19-16257-f001:**
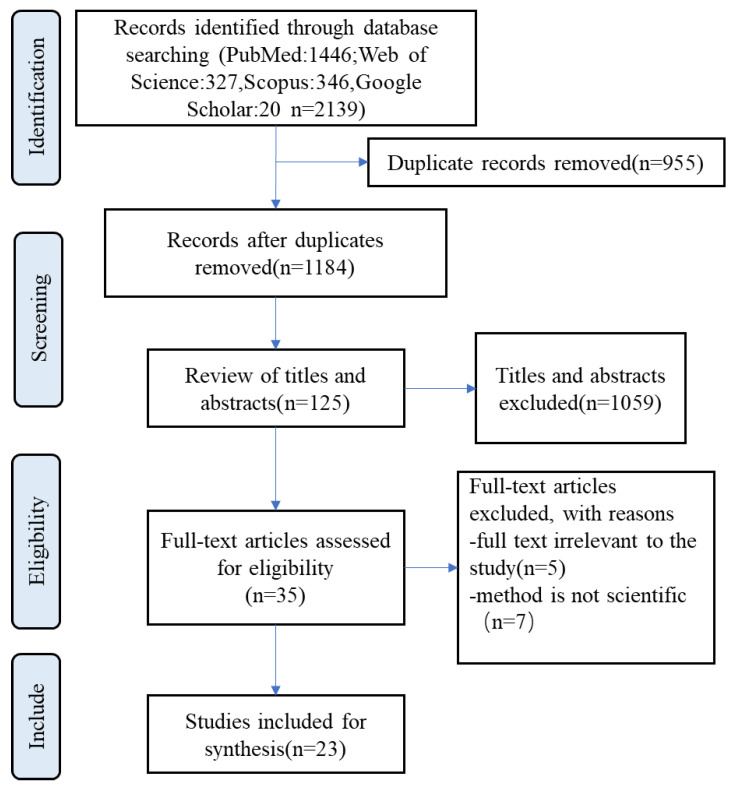
Selection process flow diagram.

**Table 1 ijerph-19-16257-t001:** Systematic Search Strategy.

	Search Terms		
	PubMed	Web of Science	Scopus
1. Set of entry criteria		TS = (hospital preparedness) OR TS = (emergency trauma)	TITLE-ABS-KEY (“bioterrorism prevention”) OR TITLE-ABS-KEY (“bioterrorism control”) OR TITLE-ABS-KEY (biological)
2. Set of entry criteria		TS = (“bioterrorism”) OR TS = (terrorism, biological) OR TS = (biological terrorism)	TITLE-ABS-KEY (hospital) OR TITLE-ABS-KEY (“emergency trauma”) OR
3. Set of entry criteria		TS = (emergency) OR TS = (emergency preparedness) OR TS = (disaster planning)	TITLE-ABS-KEY (disaster planning) OR TITLE-ABS-KEY (disaster preparedness)
Final search	(“bioterrorism/prevention and control” [Mesh]) OR (“bioterrorism/organization and administration” [Mesh]) AND/OR (“emergency service, hospital/organization and administration” [Mesh]) OR (“trauma centers/organization and administration” [Mesh])	1 AND 2 AND 3	1 AND 2 OR 3

**Table 2 ijerph-19-16257-t002:** Inclusion of exclusion criteria.

Inclusion Criteria	Exclusion Criteria
1. Study the hospital’s ability to respond to bioterrorism incidents by investigating or constructing indicators; including a review of events that have occurred and work processes.	1. The research direction consists of biological warfare agents, pathogens, etc., that are not related to management.
	2. The content involves prevention and response methods, tools, and legislation related to bioterrorism, which are not focused on hospitals.
	3. The content focuses on the quality and delivery of medical staff surveys, education, and training.
2. Any type of studies (observational, cross-sectional, longitudinal).	4. Reviews, letters, communications, notes, editorials, and conference reports.
	5. Abstract missing; full text not found.
3. Studies published in English.	6. Not published in English.
4. Studies published from 2012 to 2022.	7. Studies published before January 2012 or after September 2022.

**Table 3 ijerph-19-16257-t003:** Basic information about the 23 articles included in the review.

No.	Title	Academic Journal	Publication Date	Regional Study
1	Evidence-based support for the all-hazards approach to emergency preparedness	Israel Journal of Health Policy Research	25 October 2012	Israel
2	Construction of evaluation systems for the ability to respond to bioterrorism in military hospitals	Military Medical Sciences	25 October 2012	China
3	Suggestions about the ability to respond to bioterrorism in military hospitals	Military Medical Sciences	25 February 2014	China
4	Hospital disaster preparedness in Switzerland	Swiss Medical Weekly	2 October 2014	Switzerland
5	Evaluation and Analysis of Hospital Disaster Preparedness in Jeddah	Scientific Research	10 November 2014	Saudi Arabia
6	A Comprehensive Evaluation System for Military Hospitals’ Response Capability to Bioterrorism	Cell Biochemistry and Biophysics	22 January 2015	China
7	Hospital Disaster Preparedness in Italy: a preliminary study utilizing the World Health Organization Hospital Emergency Response Evaluation Toolkit.	Minerva Anestesiologica	7 June 2016	Italy
8	Evaluation of Hospitals’ Disaster Preparedness Plans in the Holy City of Makkah (Mecca): A Cross-Sectional Observation Study	Prehospital and Disaster Medicine	22 June 2016	Saudi Arabia
9	Are Dutch Hospitals Prepared for Chemical, Biological or Radio Nuclear Incidents? A Survey Study	Prehospital and Disaster Medicine	9 December 2016	Netherlands
10	Survey of Biological Incidents Preparedness of Hospitals in Markazi Province in 2016	Journal of Military Medicine	24 June 2017	Iran
11	How is training hospitals of Qazvin preparedness against disaster in 2015	Annals of Tropical Medicine and Parasitology	September 2017	Iran
12	A Study of Hospital Disaster Preparedness in South Yemen	Prehospital and Disaster Medicine	April 2018	Yemen
13	Hospital management preparedness tools in biological events: A scoping review	Journal of Education and Health Promotion	29 September 2019	Iran
14	Evaluation of hospital disaster preparedness by a multi-criteria decision-making approach: The case of Turkish hospitals	International Journal of Disaster Risk Reduction	22 June 2020	Turkey
15	Factors affecting hospital response in biological disasters: A qualitative study	Medical Journal of the Islamic Republic of Iran	16 March 2020	Iran
16	Hospital Preparedness Challenges in Biological Disasters: A Qualitative Study	Disaster Medicine and Public Health Preparedness	5 November 2020	Iran
17	Assessing the preparedness of hospitals facing disasters using the rough set theory: guidelines for more preparedness to cope with the COVID-19	International Journal of Systems Science: Operations&Logistics	24 March 2021	Iran
18	Hospital Disaster Preparedness in Iranian Province: A Cross-sectional Study Using A Standard Tool	American Journal of Disaster Medicine	6 April 2021	Iran
19	Disaster preparedness in emergency medical service agencies: A systematic review	Journal of Education and Health Promotion	30 July 2021	Iran
20	COSMIN Checklist for Systematic Reviews of the Hospital Preparedness Instruments in Biological Events	Journal of Nursing Measurement	1 December 2021	Iran
21	Investigating the level of functional preparedness of selected Tehran hospitals in the face of biological events: a focus on COVID-19	International Journal of Disaster Resilience in the Built Environment	10 January 2022	Iran
22	Chemical, Biological, Radiological, or Nuclear Response in Queensland Emergency Services: A Multisite Study	Health Security	24 May 2022	Australia
23	Establishing the Domains of a Hospital Disaster Preparedness Evaluation Tool: A Systematic Review	Prehospital and Disaster Medicine	17 July 2022	China

**Table 4 ijerph-19-16257-t004:** Main subcategories of hospital response to bioterrorism (superscript numbers correspond to the numbers in the document information table).

Main Subcategories	Contains Subcategories
Emergency management	organizational management ^2,3,16^, command and coordination ^2,6,7,12,18,22^, training and education ^1,2,3,6,7,8,12,13,14,15,16,19,22^, plan ^4,5,8,9,11,12,13,19,22^, communication systems ^5,7,8,12,13,14,15,16,19,21^, information management ^14,17^, lock down the facility ^6^, transport ^23^, vehicles ^2,5,14,19^, safety and security ^7,12,18,23^
Medical service capacity	prehospital first aid (prehospital team) ^2,3,6^, electronic medical records ^22^, patient management ^16,22^, triage ^7,12,15,16,18^, on-site disposal ^2,3,6^, personal protective equipment (PPE) ^3,8,9,13,16,19^, monitoring and warning ^2,6,12,13,15,19^, decontamination ^2,8,9,13,15,16,19,20,22^, body storage/disposal ^8,23^, morgue facilities ^23^, anti-infective therapy and vaccines ^15,19,23^, standard operating procedure (SOP) ^1,13,19^, isolation ^6,9,15^, treatment and infection control ^15^, continuity of essential services ^12,18^
Surge capacity	material supply ^6,14,19^, infrastructure and equipment ^1,2,3,5,6,11,13,14,17,19^, stockpiling ^23^, pharmaceuticals (antidotes ^9^, vaccines ^10^) ^11,14,17,22^, human resource ^4,5,7,12,13,14,18,19^, staff support policies ^13,15,16,22,23^, surge bed capacity ^4,7,12,14,17,18,21,22^, psychological counselling ^19^, additional staff ^4,13,23^, surge discharge plan ^8^
Laboratory	scientific research ^2,6,17^, surveillance system ^12,15,16^
Regional coordination	integration with local or regional systems ^5,13,22^, inter-organizational coordination ^15,19^, cooperative plan^3^, health facility networking ^5^, referral procedure ^5^, infectious disease control centers ^15^
Logistical support	Funding ^2,6,7,12,14,15,19^, legal ^19^, architecture and furnishings ^5,13,15,16^, safety of Lifeline Facilities (water ^14^, electricity ^23^) ^5,7,13,14^, location and areas ^5,14^, emergency supply kit ^3,17^, waste management ^16^, post-disaster recovery ^7,12,18,21^, logistics and management supply ^7,12,18^

Please see explanation of superscripted numbers in Appendix A.

## Data Availability

Restrictions apply to the availability of these data. Data was obtained from [third party] and are available [from the authors] with the permission of [third party].

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
