# Peer review of "The PRISMA 2020 Statement: A System Review of Hospital Preparedness for Bioterrorism Events"

_ijerph, 2022, doi:10.3390/ijerph192316257_

Round 1
Reviewer 1 Report
The paper is a review of literature following the PRISMA recommendations. I congratulate with the authors; the review is well written and definitely deserves to have space in peer-review journals. Hereafter some minor comments and feedbacks.
INTRODUCTION
The introduction is well structured, as it introduces the topic, the gap in knowledge and the proposed aim. I have some concerns regarding the aim itself, which according to the authors is “to summarize and analyze the relevant literature on hospital disaster preparedness and extracts relevant capability elements, which can provide a reference for the preparation of hospitals against bioterrorism and provide a basis for the design and development of hospital preparedness assessment indicators.”
From this paragraph it seems that the review will focus only on general hospital disaster preparedness, when in fact the search terms showed in the methodology suggest that the review is indeed focused specifically on the bioterrorism aspect. Therefore, I suggest to better refine the aim of the paper.
RESULTS
· “Out of these 23 studies, the majority (n = 16) were cross-sectional surveys and others were Delphin study (n = 4).” I believe you meant “Delphi studies”
· Table 3: can you lease clarify what is the “Division of Chinese Academy of Science?”
· I believe that table 3 and 4 can be merged to have one single table. In table 4 there is again the term “Delphin”
CONCLUSIONS
The paragraph “The results of this study can be used as a reference standard for the preparation of hospital bioterrorism, and provide a basis for developing the design of hospital preparedness assessment indicators, so as to help hospitals respond to bioterrorism incidents more actively and effectively.” Is present twice.
Author Response
Dear reviewer,
Thanks very much for taking your time to review and constructive comments on this manuscript. We have carefully considered the suggestions and tried our best to improve and made some changes in the manuscript. Accordingly, we have uploaded a copy of the original manuscript with all the changes highlighted by using the track changes mode in MS Word. The yellow part has been revised according to your comments. Appended to this letter is our point-by-point response to the comments.

Reviewer 2 Report
Dear editor,
It’s my honour to review this manuscript. Although few researchers have focused on bioterrorism preparedness in hospitals, such the topic is very important and may contribute to the study of the relationship between preparedness in hospitals and COVID-19 pandemic also.
General Comments:
By referring to the WHO checklist, it combined the characteristics of bioterrorism events with the emergency preparedness of hospitals, and carried out a literature review by using the PRISMA checklist, and put forward a framework of hospital preparedness. The tables are very detailed. The research content of this article is relatively novel and has certain reference value, which can arouse readers' interest in reading. I think the manuscript is acceptable, but there are a few problems that need to be corrected.
Specific Comments:
1. In the methods section, this review selects literatures in recent ten years, screens and analyzes it using PRISMA guidelines, and references official WHO documents to obtain elements of preparedness for bioterrorism events in hospitals. However, it does not clarify the advantages of PRISMA. It could be recommended that this be explained in the Introduction or Methods section.
2. The lack of functional monitoring systems during the current COVID-19 by multiple national health departments has shown that old theory and monitoring systems are too slow in identifying rapidly the presence of an acute biological emergency. Monitoring and warning is not just the responsibility of hospitals. In the discussion section, this point can be made and explained.
3. The quantitative results are just one short paragraph describing who published the studies that they included, and when, which are a little less. Could you please add some content to indicate the significance of quantitative analysis.
Author Response

(The authors gave the same response as above.)
